# CRISPR/Cas9-Mediated *SlATG5* Mutagenesis Reduces the Resistance of Tomato Fruit to *Botrytis cinerea*

**DOI:** 10.3390/foods12142750

**Published:** 2023-07-19

**Authors:** Yujing Li, Pan Shu, Lanting Xiang, Jiping Sheng, Lin Shen

**Affiliations:** 1College of Food Science and Nutritional Engineering, China Agricultural University, Beijing 100083, China; yujinglee820@163.com (Y.L.); sp199645cau@163.com (P.S.); xianglt@cau.edu.cn (L.X.); 2School of Agricultural Economics and Rural Development, Renmin University of China, Beijing 100872, China

**Keywords:** *SlATG5*, CRISPR/Cas9, JA/SA signal, *Botrytis cinerea*, tomato fruit

## Abstract

Tomato fruit is highly susceptible to infection by *Botrytis cinerea* (*B. cinerea*), a dominant pathogen, during storage. Recent studies have shown that autophagy is essential for plant defense against biotic and abiotic stresses. Autophagy-related gene 5 (*ATG5*) plays a key role in autophagosome completion and maturation, and is rapidly induced by *B. cinerea*, but the potential mechanisms of *ATG5* in *Solanum lycopersicum (SlATG5*) in postharvest tomato fruit resistance to *B. cinerea* remain unclear. To elucidate the role of *SlATG5* in tomato fruit resistant to *B. cinerea*, CRISPR/Cas9-mediated knockout of *SlATG5* was used in this study. The results showed that *slatg5* mutants were more vulnerable to *B. cinerea* and exhibited more severe disease symptoms and lower activities of disease-resistant enzymes, such as chitinase (CHI), *β-*1,3-glucanase (GLU), phenylalanine ammonia-lyase (PAL), and polyphenol oxidase (PPO), than the wild type (WT). Furthermore, the study observed that after inoculation with *B. cinerea*, the relative expression levels of genes related to salicylic acid (SA) signaling, such as *SlPR1*, *SlEDS1*, *SlPAD4*, and *SlNPR1*, were higher in *slatg5* mutants than in WT. Conversely, the relative expression levels of jasmonic acid (JA) signaling-related genes *SlLoxD* and *SlMYC2* were lower in *slatg5* mutants than in WT. These findings suggested that *SlATG5* positively regulated the resistance response of tomato fruit to *B. cinerea* by inhibiting the SA signaling pathway and activating the JA signaling pathway.

## 1. Introduction

Tomato (*Solanum lycopersicum*) is a widely cultivated horticultural crop worldwide, due to its rich nutritional content and its economic significance [1]. However, tomato production is often limited by diseases, especially *Botrytis cinerea* (*B. cinerea*) infection, which is responsible for reduced fruit quality and fruit decay, in the greenhouses and in open fields, from pre- to postharvest, and even in cold storage [2]. Therefore, research into the genetic mechanisms of defense against *B. cinerea* is essential for the long-term sustainability of tomato production.

Over time, plants have evolved a comprehensive and intricate disease-resistance regulatory network by utilizing various strategies to combat pathogens [3]. Pathogen invasion triggers a two-tiered innate immune system in plants, consisting of pathogen-associated molecular patterns-triggered immunity (PTI) and effector-triggered immunity (ETI) [4]. Upon initiation of PTI and ETI, a hypersensitive response (HR) is typically triggered. This response includes programmed cell death (PCD), production of reactive oxygen species (ROS), and synthesis of antibacterial substances at the site of infection. These mechanisms work together to prevent the spread of infection to other parts of the plant [5]. In addition, plants exhibit a rapid response to pathogens by initiating various defense mechanisms, including modification of their cell walls, accumulation of secondary metabolites, and synthesis of hormone-signaling molecules such as salicylic acid (SA) and jasmonic acid (JA) [6]. The activation of plant immune responses is regulated by SA and JA, either alone or in combination, depending on the type of pathogen detected [7,8]. The JA pathway is thought to play a positive role in defense against necrotrophic pathogens such as *B. cinerea*. The SA pathway plays an active role against non-necrotrophic pathogens, whereas necrotrophic pathogens utilize SA to promote disease development [9]. In general, SA and JA are mutually antagonistic, but there is also crosstalk and synergy between them, which together provide plants with selectable immune modes and adjustable immune strength [10].

Autophagy is an evolutionarily highly conserved cyclic process that plays a major role in plant growth, development, and stress response [11]. The process of autophagosome formation involves several steps, including signal transduction (induction), isolation membrane formation (nucleation), isolation membrane elongation (elongation), autophagosome completion and transport (maturation), autophagosome-vacuole coupling and fusion (fusion), cargo degradation (degradation), and recycling [12]. Several autophagy-related (ATG) proteins are involved in this process, among which the combination of ATG8 and phosphatidylethanolamine (PE) is critical for autophagy formation. The ATG5-ATG12 complex promotes the combination of ATG8 and PE, thereby affecting autophagy completion [13].

Autophagy has been shown to enhance plant resistance to a variety of pathogens through different mechanisms [14]. In *Arabidopsis*, infection with *B. cinerea* induced the relative expression of *ATG* genes, and the *atg5/7/18a* mutations impaired resistance to *B. cinerea* by regulating the expression of genes related to SA/JA signaling. Autophagy, together with JA- and *WRKY33*-mediated signaling pathways, regulated plant resistance responses to necrotrophic pathogens [15]. SA- and NPR1-dependent immune responses were found to be affected in *atg2 and atg5* mutants in *Arabidopsis* [16]. Furthermore, *ATG* genes play a role in tomato defense against biotic and abiotic stresses. The silencing of *SlATGs* genes led to an increased accumulation of ubiquitinated proteins, resulting in impaired brassinolide-induced cold tolerance in tomato plants [17]. The virus-induced gene silencing of tomato *SlATG5* and *SlATG7* genes contributed to increased heat stress sensitivity in tomato plants [18]. Moreover, methyl jasmonate (MeJA) treatment increased the activity of postharvest autophagy activity and subsequently enhanced MeJA-mediated anti-*B. cinerea* capacity by modulating ROS metabolism in tomato fruit [19]. However, the underlying mechanism by which autophagy contributes to tomato fruit resistance to *B. cinerea* is still unclear.

In this article, the crucial role of *SlATG5* in tomato fruit resistance to *B. cinerea* was investigated, and the mechanism was explored based on the classic disease-resistance pathway. To obtain the *slatg5* mutants, we used the clustered regularly interspaced short palindromic repeats (CRISPR)/CRISPR-associated protein 9 (CRISPR/Cas9) system, and the obtained knockout mutants of *SlATG5* were confirmed by sequencing and TEM electron microscopy. Meanwhile, we analyzed the expression profile of *SlATG5* in different organs, at different stages of maturity and under *B. cinerea* infection. The phenotypic observation and disease-resistance enzyme activity assay were studied to explore the function of *SlATG5* in tomato fruit resistance against *B. cinerea*. Additionally, the expressions of SA- and JA-related genes were examined to find out the possible regulatory mechanism of *SlATG5* in disease resistance.

## 2. Materials and Methods

### 2.1. Plant and Fruit Materials 

The tomato plants used in this experiment belong to the genus *Solanum lycopersicum* cv. Ailsa Craig background. The seeds were sown into plastic basins with a diameter of 7 cm filled with a mixture of vermiculite, soil, and seedling substrate (*v*/*v*/*v* = 1:1:2). The plants were incubated for 6 weeks at 23–25 °C, with a photoperiod of 16/8 h light/dark, and a relative humidity (RH) of 60–65%.

At the mature green stage (40 d after flowering), tomato fruit of *slatg5* mutants and wild type (WT) were harvested from a greenhouse vegetable base in Shangzhuang, Beijing, China. Only fruit with regular shape and size, no physical damage, and no pathogen infection were selected for testing. The selected fruit were immediately preserved at 20 ± 1 °C for 12 h to eliminate field heat, transported to the laboratory, washed with 3% (*v*/*v*) solution of sodium hypochlorite for 2 min, rinsed twice with distilled water, and finally air dried.

### 2.2. Material Treatments for Analysis of Expression Patterns of SlATG5

Tomato root, stem, leaf, flower, and fruit were used to determine the expression pattern of *SlATG5* in different organs. The root, stem, and leaf samples were collected from 4-week-old WT tomato plants, and the flower samples were collected from WT tomato petals at the full flowering stage (BBCH growth stage 65) [20,21], and the fruit used for analysis of expression patterns in Figure 1A,C were collected from the WT tomato fruit at the mature green stage (40 d after flowering, BBCH growth stage 73–79). Tomato samples were collected in triplicate, snap frozen in liquid nitrogen, and stored at −80 °C. Quantitative real-time PCR (qRT-PCR) was used to detect *SlATG5* expression patterns at different ripening stages, and at 0 h, 2 h, 4 h, 6 h, 12 h, and 24 h after *B. cinerea* infection. The procedure of qRT-PCR is detailed in Section 2.9 and the specific primers for *SlATG5* are listed in Appendix A.

According to the standard GH/T 1193-2021 [22] and USDA 1975 [23], tomato fruit ripening stages were classified as follows: (1) mature green (MG), where the surface of tomato fruit is entirely green, ranging from light green to dark green, BBCH growth stage 73–79; (2) breaker (B), where there is a noticeable change in color from green to tan, pink, or red, covering no more than 10% of the peels, BBCH growth stage 81; (3) turning (T), where 10–30% of the surface is not green and varies significantly in color from green to tan, pink, red, or a multiple color combination, BBCH growth stage 82–83; (4) pink (P), where 30–60% of the surface is not green and generally appears pink or red, BBCH growth stage 84–86; (5) red (R), where more than 90% of the surface is red, BBCH growth stage 89.

### 2.3. Selection of Target Sequences and Vector Construction

In previous studies, we followed a similar method for the construction of vectors [24]. To generate *slatg5* mutants, we designed two target sequences for the *SlATG5* gene utilizing the CRISPR-P 2.0 program (http://crispr.hzau.edu.cn/cgi-bin/CRISPR2/SCORE (accessed on 18 July 2023)). The target single-guide RNA (sgRNA) expression cassettes were generated by using two-step overlapping PCR. We used primers U-F, A5AtU3bT1^−^ (or A5AtU3dT2^−^), A5gRT1^+^ (or A5gRT2^+^), and gR-R for the first round of PCR. In addition, we used appropriate site-specific primers Pps-GGL/Pgs-GG2 (for target 1) and Pps-GG2/Pgs-GGR (for target 2) containing BsaI-cutting sites for Golden Gate ligation in the second round. The two sgRNA expression cassettes were then ligated into the pYLCRISPR/Cas9Pubi-H vector. Appendix A lists the primers used to construct the recombinant pYLCRISPR/Cas9 vector.

### 2.4. Agrobacterium-Mediated Transformation

The identified pYLCRISPR/Cas9Pubi-H-*SlATG5* vector was electrotranformed into *Agrobacterium* competent cells EHA105, as reported previously [25]. 

The genetic transformation of tomato was achieved using the Agrobacterium-mediated leaf disc method. The experimental material used was the wild-type Ailsa Craig (AC) tomato. Initially, the tip of the tomato cotyledon was removed, and the remaining portion was cut into small squares measuring 5 mm × 5 mm. These explants were then placed upside down on the culture medium, with the back of the leaves facing upwards. The explants were evenly arranged, sealed, and kept in a dark room at a temperature of 25 °C for pre-cultivation for a duration of two days. Subsequently, the tomato leaves were infected with *Agrobacterium* containing the reporter gene, and further cultured in a dark room at 25 °C for another two days. Following the subsequent periods of bud induction, bud elongation, rooting, and soil culture, the *atg5* transgenic plants were ultimately generated.

The transgenic lines were selected for their resistance to hygromycin. After explants were cultured in vitro, all hygromycin-positive plants (T0 transgenic plants) were transplanted into soil and planted at 25 °C under conditions with a photoperiod of 16/8 h light/dark. After the fruit of T0 plants ripened, the seeds were harvested and replanted in the soil and documented as T1 generation plants.

### 2.5. Mutation Identification

We extracted the genomic DNA from quick-frozen leaves (80–100 mg) by using the DNA quick Plant System Kit (TIANGEN Biotech Co., Ltd., Beijing, China). The total DNA of T0 and T1 transgenic plants was amplificated by using the hygromycin specific primers Hyg for and Hyg rev (Appendix A). Agarose gel (1%) electrophoresis was used to detect the target band of PCR amplification products and the size of the expected product was about 750 bp. Total DNA from hygromycin-positive plants was used as the template to amplify Target 1 by using primers ATG5-T1-F and ATG5-T1-R or Target 2 by using primers ATG5-T2-F and ATG5-T2-R. The PCR program was first at 94 °C for 3 min, followed by 35 cycles of 94 °C for 30 s, 55 °C for 30 s, and 72 °C for 30 s, and 72 °C for 7 min. Finally, PCR products were subjected to Sanger sequencing with primer T1/T2 seq (Appendix A). Overlapping sequence chromatograms were decoded for analysis by using DSDecodeM (http://skl.scau.edu.cn/dsdecode/ (accessed on 18 July 2023)).

### 2.6. Pathogen Inoculation Treatment and Determination of Disease Symptoms

*B. cinerea* was inoculated into fruit, as described in our previous report [26,27]. Briefly, *B. cinerea* (ACCC 36028) was obtained from the China Agricultural Culture Collection Center in Beijing, China, inoculated on potato dextrose agar (PDA) medium, and cultured at 25 ± 1 °C in the dark for 10 d to promote sporangia production. Spores were gathered by gently scraping the medium surface, resuspended in sterile distilled water, and prepared as 2 × 10^5^ CFU mL^−1^ spore suspension.

Two symmetrical holes (4 mm deep × 2 mm wide) were punched in the equatorial part of each fruit, and 10 µL of spore suspension was inoculated into each hole by using a sterile pipette tip. Fruit inoculated with pathogens were stored at 25 ± 1 °C with 85–90% RH to induce disease. Twenty fruits from each group were selected for phenotypic observation and lesion diameter measurement. Samples were collected from five fruits at 0 h, 2 h, 4 h, 6 h, 12 h, 24 h, 48 h, and 96 h after inoculation for enzyme activity assay and gene expression. The mesocarp tissues without all seeds were collected from the equatorial region of the fruit after removal of the lesion area, cut into pieces, flash frozen in liquid nitrogen, ground to a fine powder with a grinder (IKA-Werke GmbH, Staufen, Germany), and processed into frozen fruit samples, which were stored at −80 °C. On the third and fifth days after inoculation, photographs were taken to record the lesion, and the diameter of the lesion was measured.

### 2.7. The Transmission Electron Microscopy (TEM) Analysis

On the second day after inoculation with *B. cinerea*, a small square (about 5 mm^2^) was taken from the edge of the lesion of the fruit mesocarp, using a scalpel. The samples were processed according to [28,29] with some modifications. The mesocarp samples were rapidly soaked in the E-64 d (loxistatin) solution for 24 h to prevent the degradation of the formed autophagosomes, for easy observation. The samples were then fixed in 2.5% (*v*/*v*) glutaraldehyde for at least 48 h at room temperature in the dark and washed three times with 0.1 M phosphate buffer saline (PBS, pH 7.2). The samples were then fixed with 1% (*w*/*v*) osmic acid for 3 h and washed three times with 0.1 M phosphate buffer (PBS, pH 7.2), then dehydrated through graded ethanol solutions (30, 50, 70, 90, and 100%), washed with propylene oxide, and embedded in Spurr resin at 60 °C for 24 h. The embedded samples were then cut into ultrathin slices of 80 nm using an ultramicrotome. Finally, the fruit samples were observed with a transmission electron microscope (Hitachi Ltd, Tokyo, Japan) and photographs were taken simultaneously.

### 2.8. Activities of Defense Enzymes

A 0.5 g frozen fruit sample was added to 5 mL of 0.2 mmol L^−1^ boric acid buffer at pH 8.8 containing 10% (*w*/*v*) polyvinylpyrrolidone (PVP) and 1 mmol L^−1^ EDTA for phenylalanine ammonia lyase (PAL) activity assay; to 5 mL of 100 mmol L^−1^ acetic acid buffer at pH 5.2 for chitinase (CHI) and *β*-1,3-glucanases (GLU) activity assays; and to 5 mL of PBS buffer at pH 6.8 for polyphenol oxidase (PPO) activity assays. After homogenization, samples were centrifuged at 12,000× *g* for 20 min at 4 °C, the pellet was discarded, and the supernatant was collected. Enzyme activity assays for PAL, CHI, GLU, and PPO were described in detail in our previous studies [30,31,32]. The results were expressed on a fresh weight basis, and the PAL, CHI, GLU, and PPO activity were expressed in unit (U) kg^−1^.

### 2.9. Quantitative Real-Time PCR (qRT-PCR)

Total RNA was extracted from 0.2 g of tomato fruit of each group using the EasyPure Plant RNA Kit (Beijing Transgen Biotech Co. Ltd., Beijing, China). The extracted RNA was reverse transcribed into cDNA using TransScript One-step gDNA Removal and cDNA Synthesis SuperMix (Beijing Transgen Biotech Co. Ltd., Beijing, China). All qRT-PCR were performed with TransStart Top Green qPCR SuperMix (Beijing Transgen Biotech Co. Ltd., Beijing, China) in the Bio-Rad PCR System CFX96 (Bio-Rad, Hercules, California, USA). One μL of TransStart Top Green qPCR SuperMix, 0.3 μL of specific primer (Appendix A), and 1 μL of cDNA were added to a 10 μL final volume per reaction. The qRT-PCR procedure was set to run 40 cycles of 95 °C for 30 s, 60 °C for 15 s, and 72 °C for 15 s immediately after 95 °C for 2 min. The cycle threshold (Ct) values were used to represent the relative expression levels of the tested genes in tomato plants, normalized to the reference gene *β*-actin (accession number: AB695290.1), and calculated by using the formula 2^−ΔΔCt^. The specific primers for the detected genes are listed in Appendix A.

### 2.10. Statistical Analysis

SPSS 25.0 software (IBM Corporation, Armonk, NY, USA) was used for statistical analysis of the data. All data are expressed as mean ± standard deviation (SD). Differences were determined by Student’s *t*-test. Single (*) and double (**) asterisks indicate significant differences from WT at *p* < 0.01 and *p* < 0.05, respectively.

## 3. Results

### 3.1. Analysis of Expression Patterns of SlATG5 

To investigate the tissue specificity of *SlATG5* expression, we measured its relative expression in different organs. Our results revealed that *SlATG5* was expressed in all organs tested, with particularly higher expression levels in roots and fruit (Figure 1A). We also examined the expression of *SlATG5* at different fruit-ripening stages and found that its expression level was the highest at the turning stage, followed by the mature green stage, and relatively low at the breaker, pink, and red ripening stages (Figure 1B). The results demonstrate the potential of studying the gene function of *SlATG5* using the tomato fruit of the mature green stage, and that *SlATG5* may have a crucial effect on the growth and development of tomato. In addition, we observed that the expression of *SlATG5* was upregulated at 2 h after *B. cinerea* infection and then fluctuated, indicating that *B. cinerea* infection activated the expression of *SlATG5* and *SlATG5* played a role in the anti-*B. cinerea* response of tomato fruit (Figure 1C).

### 3.2. Generation of slatg5 Mutants by Using the CRISPR/Cas9 Gene-Editing System

The two target sequences selected for CRISPR/Cas9 editing were located in the second and fifth exons, which encode important functional domains of the SlATG5 protein (Figure 2A). The resulting vector was then used to generate transgenic tomato plants, as illustrated in Figure 2B.

Twenty-two hygromycin-resistant tomato plants were obtained by *Agrobacterium*-mediated transformation. Among them, the target gene mutation occurred in eight transgenic plants, resulting in an editing efficiency of 36.36%. Among the eight T0 plants, there were five heterozygous mutations, three biallelic mutations, and no homozygous or chimeric mutations. Appendix A shows the representative mutations in the T0 generation.

To investigate the heritability of the knockout of *SlATG5* generated by the CRISPR/Cas9 system, three transgenic lines (L2, L3, and L7) were selfed, and T1 generation plants were genotyped (Appendix A). The results showed that the editing patterns of the transgenic plants were different in the T1 generation. Plants of the L3 line were restored to the WT. Line 2 contained heterozygotes and biallelic mutants with large deletions, resulting in a single amino acid substitution in the former and a shorter, non-functional protein in the latter. Line 7 was a biallelic mutant with a deletion of 14 bases, resulting in an apparent frameshift mutation and loss of protein function. Finally, line 2 was selected for the following experiments.

### 3.3. Knockout of SlATG5 Blocked Formation of Autophagosomes

Detection of autophagy by TEM analysis of autophagosomes is a critical benchmark. Once autophagy is activated, the double membrane structure of autophagosomes engulfs substances such as organelles, proteins, and other macromolecules. These autophagosomes are then transported to the vacuole, where the hydrolase degrades the inner plasma membrane of the autophagosome, and the degradation products are transported to the cytoplasm for reuse by the cell.

It was observed that a considerable amount of autophagosomes were present in the WT pericarp cells indicated by the red arrow in Figure 2D, including autophagosomes that had formed double-membrane structures and autophagic vesicles with single-layer membrane structures that may have undergone partial degradation. However, the pericarp cells of the *slatg5* mutants exhibited a large number of incomplete autophagic vesicle structures, with poorly formed autophagic bilayer membranes and no fully formed autophagosomes detected, as indicated by the yellow arrow in Figure 2E. Our results indicated that knockout of *SlATG5* disrupted the expansion of autophagosomal membranes and the formation of autophagosomes, which was consistent with the disruption of autophagy in *atg5* mutants observed in other studies [18,33].

### 3.4. Knockout of SlATG5 Reduced Tomato Fruit Resistance against B. cinerea

To explore the effect of *SlATG5* on the tomato fruit tolerance to *B. cinerea*, we observed the lesion phenotype of WT and *slatg5* mutants at three and five days after inoculation with *B. cinerea*. Due to the extremely low survival rate of homozygous *slatg5* mutants, we used the biallelic mutants in this study. Figure 2C shows the editing mode of *slatg5*. At 3 d and 5 d after inoculation, both WT and *slatg5* mutants showed obvious disease symptoms, including lesion expansion and growth of the *Botrytis* mycelium. However, the disease phenotype of the *slatg5* mutants was significantly more severe than that of the WT (Figure 3A). Furthermore, the statistics and lesion diameter values shown in Figure 3B,C illustrated that the lesion diameters of the *slatg5* mutants were 39.53% and 35.28% higher than that of WT at 3 d and 5 d after inoculation, respectively (*p* < 0.01). These results demonstrated that *slatg5* mutants were more susceptible to *B. cinerea* and the knockout of *SlATG5* reduced the resistance of tomato fruit to *B. cinerea*.

### 3.5. Knockout of SlATG5 Decreased Activities of Defense Enzymes

To further confirm the increased susceptibility of *slatg5* mutants to *B. cinerea*, the activities of defense enzymes such as CHI, GLU, PAL, and PPO were determined at 0 d, 1 d, 2 d, 3 d, 4 d, and 5 d after inoculation. Figure 4A illustrates that the activities of CHI in both WT and *slatg5* mutants showed a fluctuating upward trend. However, CHI was 10.65% less active in *slatg5* mutants than WT at 2 d, showing a significant difference. The activity of GLU in the WT and *slatg5* showed an upward trend before decreasing with a peak at 2 d post-inoculation. From the second day after inoculation, GLU enzyme activity started to decrease in both *slatg5* mutants and WT. Moreover, the GLU enzyme activity in the *slatg5* mutants was significantly lower than that in the WT (Figure 4B). In addition, as shown in Figure 4C, the activities of PAL were 58.38% and 31.58% lower in *slatg5* mutants than in WT at 2 d and 4 d, respectively, showing extremely significant differences at 2 d. Regarding PPO enzyme activity, it was observed that it initially increased and then decreased in both *slatg5* mutants and WT after inoculation. The highest activity was observed at 2 d. However, PPO enzyme activity was always significantly lower in *slatg5* mutants compared to the WT after inoculation (Figure 4D). These results demonstrated that knockout of *SlATG5* reduced the enzymatic activities of CHI, GLU, PAL, and PPO to different degrees, further confirming the positive role of *SlATG5* in tomato fruit resistance to *B. cinerea*.

### 3.6. SlATG5 Regulated the Expression of Defense-Related Genes 

The involvement of the JA and SA signaling pathways in the defense response of plants is crucial and cannot be overlooked in the investigation of plant disease resistance. To elucidate the mechanism of *SlATG5*-mediated regulation of *B. cinerea* resistance, the expression levels of defense genes related to the SA and JA pathways were examined at multiple time points during the first day after infection. The examined genes included two essential SA biosynthetic genes, *EDS1* (enhanced disease susceptibility 1) and *PAD4* (phytoalexin deficient 4), an SA-related marker gene *PR1* (pathogenesis-related 1), an SA signal receptor gene *NPR1*, a JA biosynthetic gene *LoxD* (lipoxygenase), a transcriptional regulator for several JA-responsive genes *MYC2*, and a negative regulator of the JA signaling pathway *JAZ1* (jasmonate zim domain 1). The results showed that the expression levels of *SlEDS1*, *SlPAD4,* and *SlPR1* in WT decreased over time after inoculation, and the expression level of *SlNPR1* showed an initial increase followed by fluctuations. Additionally, the expression levels of these genes in WT were significantly lower than those in *slatg5* mutants at most time points post-inoculation. Notably, the relative expression levels of *SlEDS1*, *SlPAD4*, and *SlNPR1* in *slatg5* mutants peaked at 12 h, and the expression level of *SlPR1* peaked at 24 h, which were 5.12, 5.07, 6.78, and 7.89 times higher than those found in the WT, respectively (Figure 5A–D). In addition, the relative expression levels of *SlLoxD* and *SlMYC2* were upregulated in the WT after inoculation, while the expression levels of *SlLoxD* and *SlMYC2* in *slatg5* mutants did not change significantly and were lower than those in the WT throughout the experimental course (Figure 5E,F). The expression of *SlJAZ1* in WT and *slatg5* plants varied over time. However, in *slatg5* mutants, it was higher at 2 h and 6 h and lower at other time points than that in the WT (Figure 5G). These results suggested that the SA signaling pathway was upregulated, while the JA signaling pathway was downregulated upon infection with *B. cinerea* after the knockout of *SlATG5*, ultimately leading to decreased disease resistance of tomato fruit.

## 4. Discussion

Autophagy is a well-conserved dynamic biological process that enables the recycling of intracellular materials through the degradation of cytosolic molecules [12,34]. In eukaryotes, autophagy plays a role in various stages of plant growth and development, especially in response to environmental stress and pathogen infection, by maintaining the stability of the intracellular environment [35,36]. More than 30 *SlATG* genes have been identified in tomato, and the complex formed by ATG5 and ATG12 controls the association of ATG8 with PE, a key milestone step in autophagosome formation [37]. Phenotypic and autophagic flux analysis of *atg* mutants revealed that *atg7*, *atg10*, *atg5*, and *atg2* mutants exhibited complete or nearly complete inhibition of autophagy in *Arabidopsis* [33]. However, studies on the importance of *SlATG5* in the response of tomato fruit to biotic stresses remain limited. Investigating the role of *SlATG5* in the response to *B. cinerea* in tomato fruit will not only help to deepen the understanding of the mechanism of *SlATG5*, but also provide support for breeding new cultivars that are more adaptable to changing environments.

In this research, we analyzed the relative expression of *SlATG5* in different organs and developmental stages of tomato. The results revealed that the transcript level of *SlATG5* was higher in roots and fruit than in other organs, with the highest level observed at the turning stage, followed by the mature green, breaker, pink, and red ripe stages (Figure 1A,B). After infection, *SlATG5* expression was upregulated (Figure 1C), indicating its involvement in the response to *B. cinerea*, consistent with the increased transcript level of *ATG5* in *Arabidopsis* after inoculation with *B. cinerea* [15].

We generated *slatg5* mutants using CRISPR-Cas9 technology and used the mutants to study the tomato fruit’s resistance to *B. cinerea*. The edited T0 alleles were heritable. However, their propagation did not follow the classical laws of inheritance, suggesting that most of the mutations in the T0 generation occurred somatically. This was consistent with studies in rice and *Arabidopsis* that indicated that most mutations in early generations were somatic [38,39]. In addition, some novel editing types were transmitted from T0 heterozygous lines carrying WT alleles to T1 generations, and similar results were found in *Arabidopsis* [40]. Biallelic mutants of line 2 had large deletions that resulted in truncated and nonfunctional translated proteins. Therefore, lines with this genotype were selected for the following experiments, and each plant was sequenced prior to the experiment.

The complex of ATG5 and ATG12 is essential for autophagosomal membrane expansion and autophagosome maturation. The TEM observation results showed that autophagosomes gathered around the lesion in WT tomato fruit upon inoculation, while the membrane structure of autophagosomes was impaired in *slatg5* mutants, indicating that the knockout of *SlATG5* affected the formation of autophagosomes (Figure 2D,E).

In addition, the phenotype analysis and lesion statistics indicated that knockout of *SlATG5* weakened the disease resistance to *B. cinerea* and led to a significant increase in the lesion area of tomato fruit (Figure 3). CHI, GLU, PAL, and PPO are the major antifungal enzymes. CHI and GLU pertain to the PR-3 and PR-2 families, respectively, and play important roles in fungal cell wall degradation [41,42]. The *slatg5* mutants had significantly lower GLU enzyme activities from 2 d after inoculation with *B. cinerea* (Figure 4A,B) and showed significantly lower CHI enzyme activities on the second day, which contributed to the reduced resistance (Figure 3). PAL is the key and rate-limiting enzyme of the phenylpropanoid metabolic pathway, and the products of the phenylpropanoid metabolic pathway can help plants effectively resist the infection by pathogens [43]. PPO is an enzyme that plays a vital role in the polymerization of phenols, which affects the synthesis of lignin in cell walls. When plants are infected with pathogens, phenolic substances can accumulate in plants. The reaction of phenolics with PPO produces quinone, which can cross-link with other substances and act as a physical barrier to protect plant tissues [44]. These enzymes work synergistically to regulate the process of disease resistance in tomato fruit. This study found that knockout of *SlATG5* resulted in lower GLU and PPO activities after *B. cinerea* infection (Figure 4C,D), which in turn impaired tomato fruit resistance to *B. cinerea* (Figure 3). The activities of other defense enzymes, such as PAL and CHI, were also reduced to varying degrees. Taken together, the results showed that knockout of *SlATG5* reduced the activities of several defense enzymes, further suggesting that *SlATG5* plays an important role in enhancing tomato fruit resistance to *B. cinerea*.

Furthermore, our results showed that knockout of *SlATG5* in tomato fruit increased the relative expression of *SlEDS1* and *SlPAD4*, which are SA synthesis genes, and the SA-related marker gene *SlPR1* after *B. cinerea* inoculation, indicating that knockout of *SlATG5* activated SA signaling after infection (Figure 5A–C), whereas the relative expression of the JA synthesis gene *SlLoxD* and the JA regulator *SlMYC2* [45,46] was lower in the *slatg5* mutants than in the WT after inoculation (Figure 5E,F), and the expression level of the JA-negative regulator *SlJAZ1* varied at different time points (Figure 5G). This indicated that knockout of *SlATG5* reduced the synthesis and transduction of JA signal.

When pathogens infect plants, the SA and JA pathways are two well-known disease-resistant pathways. These pathways can function independently, depending on the type of pathogen. It is commonly understood that JA is particularly effective against necrotrophic pathogens, while SA plays a significant role in combating non-necrotrophic pathogens [6,7,8].

Due to the complexity of the disease resistance network, there is crosstalk between SA and JA, which jointly regulate the process of plant disease resistance through synergy or antagonism [10,47]. This crosstalk is concentration-dependent, with SA and JA being mutually antagonistic at high concentrations. When tomato plants are infected by *B. cinerea*, the pathogen produces extracellular polysaccharides that act as elicitors of the SA pathway, promoting SA accumulation. In addition, the antagonism of the JA pathway by *NPR1* leads to pathogenicity in tomato [48]. However, some studies have also found that under certain circumstances, JA can induce the synthesis of SA through *MYC2*, supporting the accumulation of SA. This mechanism helps minimize the adverse effects on plant adaptability and provides plants with elastic immunity [49].

The study found that knocking out *SlATG5* increased the synthesis and activation of SA while inhibiting the synthesis and downstream transcription factors of JA. These results suggested that *SlATG5* regulated the mutual antagonism between SA and JA in tomato fruit resistance to *B. cinerea*. This was consistent with previous findings that *B. cinerea* used SA to inhibit JA and enhanced its virulence [9]. Additionally, the mechanism by which *Arabidopsis ATG5* deletion activated *PR1* and inhibited *PDF1.2* gene expression was similar. Therefore, we hypothesized that *SlATG5* suppressed the SA pathway and promoted the JA pathway to participate in the resistance of tomato fruit to *B. cinerea* [15].

To our knowledge, this is one of the few studies on the resistance to *B. cinerea* in tomato fruit by using *slatg5* mutants. Our findings provide fundamental insights into the regulatory mechanism of *SlATG5*-mediated resistance to *B. cinerea* in tomato fruit. However, since there are several autophagy related genes in tomato, further research is required to determine how *SlATG5* affects autophagy activity and how it collaborates with other autophagy genes.

## 5. Conclusions

In conclusion, knockout of *SlATG5* reduced the resistance of tomato fruit to *B. cinerea*. In addition, *SlATG5* may be involved in fruit resistance to *B. cinerea* through negative regulation of the SA signaling pathway and positive regulation of the JA signaling pathway.

## Figures and Tables

**Figure 1 foods-12-02750-f001:**
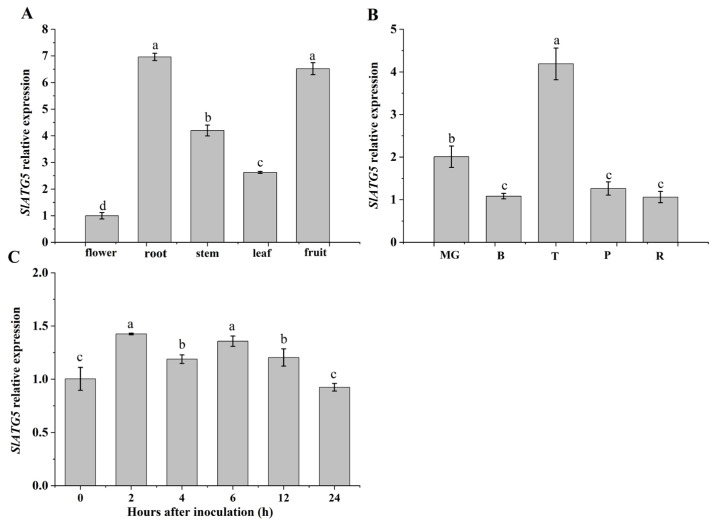
Expression patterns of *SlATG5* in different plant organs, at different maturity stages, and under *B. cinerea* infection. Relative expression of *SlATG5* in different organs of wild-type (WT) tomatoes (**A**), at different maturity stages (**B**), and under *B. cinerea* infection (**C**) of WT tomato fruit. Different letters above the histograms in the same figure reveal that the means are notably different at *p* < 0.05 after Student’s *t*-testing. Data (mean ± SD) were calculated from three biological replicates.

**Figure 2 foods-12-02750-f002:**
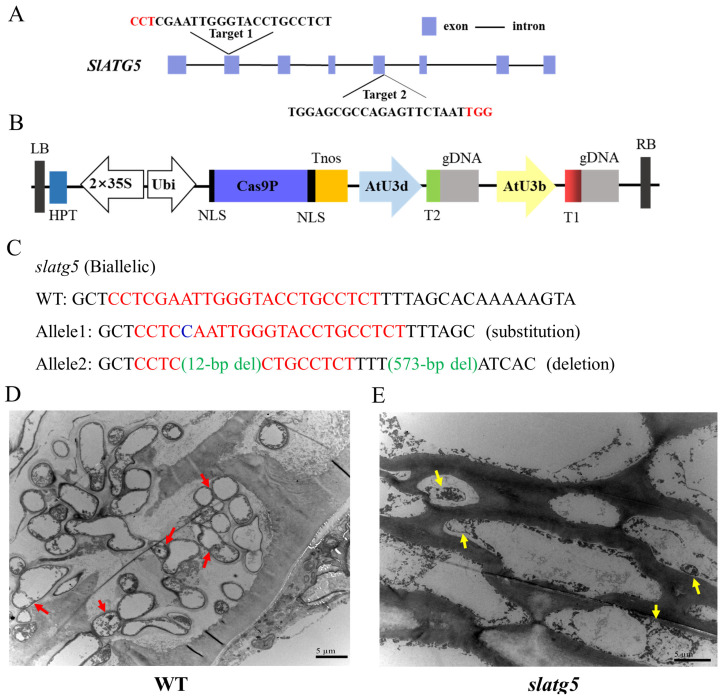
Schematic diagram of *SlATG5* gene structure and vector construction, editing type of the *slatg5* mutants, and autophagosomes in WT and *slatg5* mutants. (**A**) Schematic representation of *SlATG5* gene structure and target sequences. Purple rectangles represent exons, and black horizontal lines represent untranslated regions. Target sequences are marked with capital letters, and protospacer-adjacent motif (PAM) sequences are highlighted in red. (**B**) Schematic diagram of the pYLCRISPR/Cas9-ATG5 vector. HPT, hygromycin B phosphotransferase; Ubi, maize ubiquitin promoter; NLS, nuclear localization sequence; Tnos, gene terminator; AtU 3d, *Arabidopsis* U3d promoter; AtU3b, *Arabidopsis* U3b promoter. (**C**) Editing type of the *slatg5* mutants used in the experiment. Target sequences are marked with red letters. Deletions are indicated in green. Substitutions are indicated with blue letters. Autophagosomes in (**D**) WT tomato fruit and (**E**) *slatg5* mutants at 2 d after inoculation with *B. cinerea*. Red arrows indicate autophagosomes. Yellow arrows indicate autophagic vesicles without a complete membrane structure.

**Figure 3 foods-12-02750-f003:**
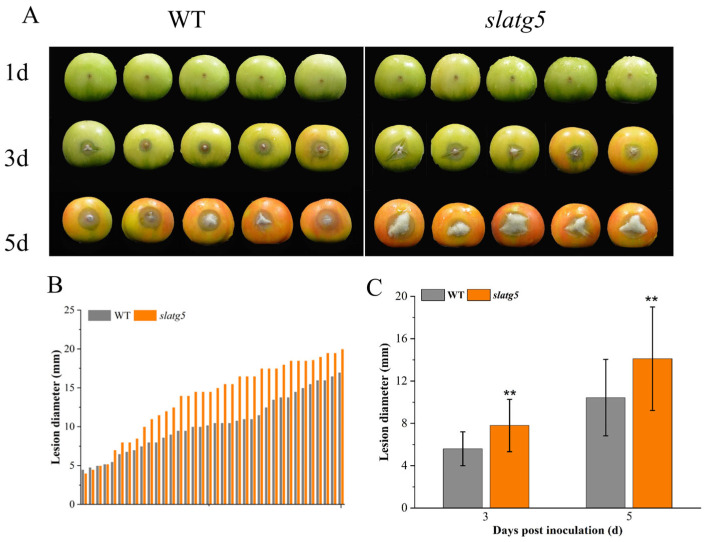
Knockout of *SlATG5* reduced the resistance of tomato fruit to *B. cinerea*. (**A**) Disease phenotype of wild type and *slatg5* fruit on 1 d, 3 d, and 5 d after inoculation with *B. cinerea*. Effect of *SlATG5* knockout on (**B**) lesion diameter distribution and (**C**) lesion diameter statistics. Asterisks represent significant differences after Student’s *t*-testing (** *p* < 0.01).

**Figure 4 foods-12-02750-f004:**
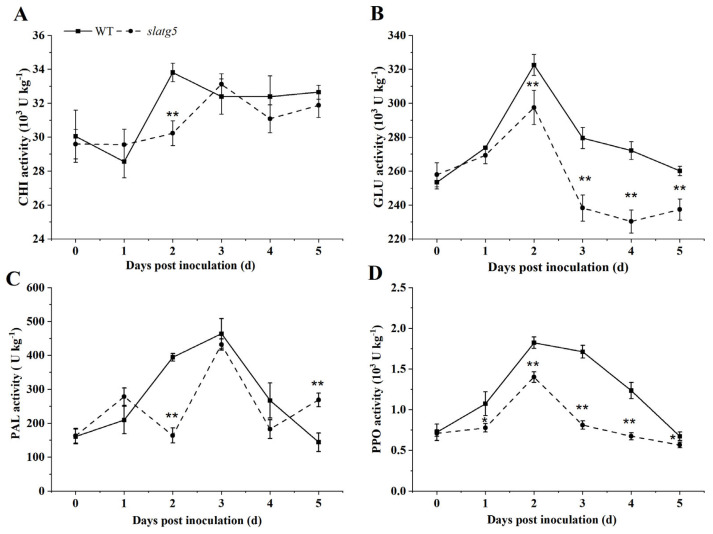
Effects of knockout of *SlATG5* on the activities of defense enzymes after inoculation with *B. cinerea*, including (**A**) chitinase (CHI), (**B**) *β*-1, 3-glucanases (GLU), (**C**) phenylalanine ammonia-lyase (PAL), and (**D**) polyphenol oxidase (PPO). Error bars indicate deviation of three biological replicates. Asterisks represent significant differences after Student’s *t*-testing (* *p* < 0.05; ** *p* < 0.01).

**Figure 5 foods-12-02750-f005:**
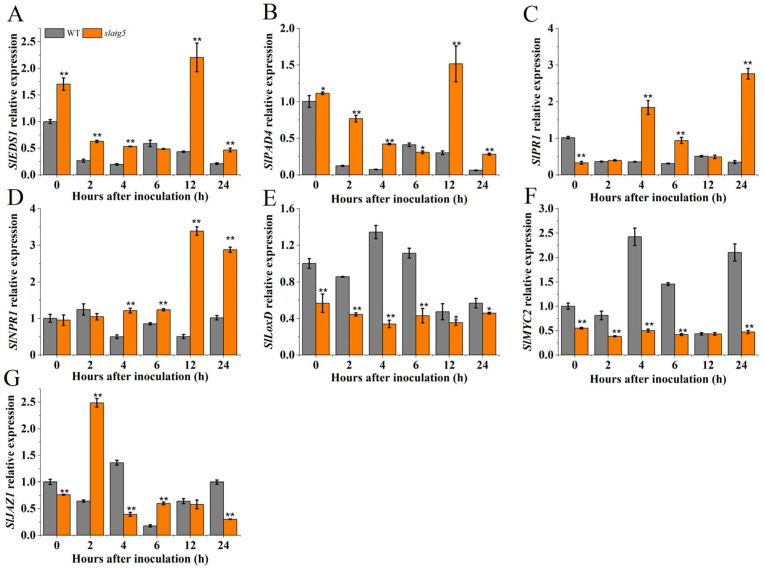
Effect of knockout of *SlATG5* on the expression levels of SA/JA signal-related genes: (**A**) *SlEDS1*, (**B**) *SlPAD4*, (**C**) *SlPR1*, (**D**) *SlNPR1*, (**E**) *SlLoxD*, (**F**) *SlMYC2*, and (**G**) *SlJAZ1*. Error bars represent standard deviation of three measurements. Significant differences between *slatg5* mutants and wild type (WT) were compared by Student’s *t*-test. Double asterisks (**) and single asterisks (*) indicate significant differences at *p* < 0.01 and *p* < 0.05.

## Data Availability

Data is contained within the article (or Appendix A).

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
