# Peer review of "CRISPR/Cas9-Mediated SlATG5 Mutagenesis Reduces the Resistance of Tomato Fruit to Botrytis cinerea"

_foods, 2023, doi:10.3390/foods12142750_

Round 1
Reviewer 1 Report
The manuscript described the role of SlATG5 in the response of tomato fruit to B. cinerea infection, through the study of slatg5 mutants obtained by CRISPR/Cas9 system. Furthermore, disease resistance enzyme activity and the expressions of SA- and JA-related genes were examined to find out possible regulatory mechanism of SlATG5 in disease resistance. The manuscript is well structured, and the proportion of chapters is appropriate. The results are well presented and discussed in detail. I would like to draw the author’s attention to some minor revisions:
- All tables in text are named as Table A1 (line 137), A2 (line 149, 155), A3 (line 211), A4 (line 248), A5 (line251) but the tables are in supplementary data and named as Table S1, etc…
- line 39: “pathegen” in place of pathogen
- line 52: “B. cinerea” in italicus
- line 103: “wide type” in place of wild type
- line 113 “tomato petals at the full flowering stage”, and line 114 “fruit of WT tomato fruit at the green ripening stage”, for greater accuracy, should be referred to BBCH-scale to identify the phenological development stage of solaneous plant and fruit.
- line 113-114: “…and the fruit used for tissue specificity and infection expression analysis was collected from the fruit of WT tomato fruit at the green 114 ripening stage approximately 40 d after flowering.” This sentence is confusing, should be reworded.
- Line 116-117: the qpcr conditions and primers used should be added.
- Line 199: insert reference for “standard GH/T 1193-2021 and USDA 1975”.
- Line 129: according to the journal's rules, only the numbers should be used in references, which should be corrected throughout the manuscript.
- In the paragraph 2.,5 the description of genetic transformation mediated by Agrobacterium is missing: what starting material was used? Were the explants obtained by direct organogenesis?
It is important to take this into account to avoid the formation of chimeras or somatic mutations (see line 391-392, in discussion).
- Line 148: how the T1 plants were obtained?
- Line 150: what is the size of the expected product?
- Line 151: total DNA is the template in PCR reaction: should be substitute “fragments” with “template”.
- Line 202- 211: the manufacturers of the kits should be added.
- In caption of figure 2 (C ) line 268: insertion indicated in green letters are missing.
Reviewer 2 Report
This is an interesting study. Botrytis cinerea, a causal agent of gray mold, is one of the main causes of postharvest fruit decay which results in considerable economic losses. To elucidate the significance of SlATG5 (an autophagy-related gene 5 in tomato) in the process of tomato fruit anti-B. cinerea, the SlATG5 gene knockout tomato was constructed by using the CRISPR/Cas9 system. After being inoculated with B.cinerea, the incidence was observed, and the activity of disease-resistant enzymes in SlATG5 and WT, the gene expression of disease-resistant signaling pathways were detected, which revealed the important role of SlATG5 in tomato fruit disease resistance. The SA and JA signaling pathways is also described in the discussion. This experiment is well designed, the data are supportive, and the results are innovative. However, there are some suggestions before it could be accepted.
1. Line 16, SlATG5, it means autophagy-related gene 5 in tomato (Solanum lycopersicum).
2. Line 34, 35, 299, 318, “greenhouse” “field” “diameter” “mutant” should be changed to plural form.
3. Line 234, 305, 364, “wildtype” should be revised to “wild-type” or “wild type”.
4. Line 287, Change "disruped" to "disrupted".
5. Pay attention to the consistency of tense. Such as line 25 “may”, line 288 “is”, line 392 “is”, line 441, “hypothesize”...
6. The mutants described in this paper are in different plants, which should be clearly described. Such as line 375“atg7,atg10,atg5,atg2”are from the Arabidopsis.
7. The enzyme activities of CHI and PAL have significant differences only in 2 d in figure 4, how to explain this phenomenon.
8. The introduction and discussion section should provide a more detailed explanation of the interplay between SA and JA signaling pathways, including their potential for synergy, crosstalk, or antagonism.
